# Acute Effect of Four Stretching Protocols on Change of Direction in U-17 Male Soccer Players

**DOI:** 10.3390/sports11090165

**Published:** 2023-09-01

**Authors:** Mohamed Amine Ltifi, Mohamed Chedly Jlid, Jérémy Coquart, Nicola Maffulli, Roland van den Tillaar, Ridha Aouadi

**Affiliations:** 1Higher Institute of Sport and Physical Education of Ksar Said, Manouba 2010, Tunisia; mohamedltifi19@gmail.com (M.A.L.); chedly3@yahoo.fr (M.C.J.); ridha_aouadi@yahoo.fr (R.A.); 2Research Laboratory (LR23JS01) “Sport Performance, Health & Society” Higher Institute of Sport and Physical Education of Ksar Said, University of Manouba, Manouba 2010, Tunisia; 3Univ. Lille, Univ. Artois, Univ. Littoral Côte d’Opale, ULR 7369-URePSSS-Unité de Recherche Pluridisciplinaire Sport Santé Société, Lille, BP 10665-62030 Arras, France; jeremy.coquart@univ-lille.fr; 4Department of Trauma and Orthopaedic Surgery, Università of Rome ‘La Sapienza’, 00185 Rome, Italy; n.maffulli@qmul.ac.uk; 5Department for Sports Science and Physical Education, Nord University, 7600 Levanger, Norway

**Keywords:** strength and conditioning, directional training, dynamic stretching, flexibility

## Abstract

Background: The ability to rapidly change direction while sprinting is a desirable athletic skill in soccer. Enhancing change of direction (COD) performance depends almost exclusively on specific training, with stretching traditionally considered one such intervention. However, the comparative impact of diverse stretching methods on COD in soccer players remains an area of interest. Therefore, this study aimed to compare the effects of different stretching methods on COD ability in soccer players. Methods: Twelve male soccer players playing in the national championship football division II (age: 16.3 ± 0.3 years, height: 1.81 ± 0.10 m, body mass: 67.7 ± 7.2 kg) were tested for COD performance (i.e., Illinois agility test) after (1) control condition (20 min general warm-up without stretching), (2) static stretching, (3) dynamic stretching, (4) combined static-dynamic stretching, and (5) combined dynamic-static stretching. The duration of stretching intervention was approximately 6 min for static and dynamic stretching and 12 min for both the combined stretching conditions. The experimental sessions were separated by 72 h. Results: COD improved after dynamic stretching when compared to any other condition (*p*: 0.03–0.002; η_p_^2^: 0.56–0.73), except for the control condition (*p* = 0.146; η_p_^2^ = 0.18). In contrast, static stretching induced a detrimental effect on COD when compared only to the dynamic stretching condition (*p* < 0.01; ES = 1.35). Conclusion: Dynamic stretching exercises used by male soccer players in the warm-up improved COD. Other forms of stretching exercises, particularly static stretching, negatively impacted the COD performance. Therefore, coaches can consider integrating dynamic stretching protocols tailored to the athletes’ specific needs. Moreover, extending the investigation to encompass a wider range of athletes, including different age groups and genders, would enhance the applicability and generalization of the findings.

## 1. Introduction

Football is a highly demanding sport, wherein players are subjected to a multitude of activities requiring substantial aerobic capacity and intermittent non-continuous exercises, which include repeated sprinting ability, agility, jumping, COD, and flexibility [1,2]. The speed and agility training method refers to a training approach based on movement tasks performed with a high rate in a short time (quickness) combined with straight (speed) and COD over a variety of distances, with and without cognitive stimuli (agility) [2]. Several authors have shown the positive effects of speed, agility, and quickness training on agility and soccer-related performance in young adults, adolescents, and preadolescents soccer players [3]. This training enhanced players’ ability to react to stimuli, accelerate, and move effectively in various directions, as well as their COD [4], which is a desirable athletic skill in soccer [5,6,7].

Time and motion analyses have evidenced that sprints with a single COD represented ~8.5% of actions in soccer [8]. During a game, soccer players accomplish around 700 direction changes of varying intensity. Approximately 600 of these changes in direction are 0–90° turns [7], and 50 of the direction changes in a soccer match are performed at maximal intensity [9]. In addition, COD ability was considered the most important performance variable for predicting soccer player selection [10], distinguishing between elite and sub-elite soccer players [11] and predicting on-field performance [12,13].

COD and other soccer-specific skills should be evaluated several times during the sport season, to assess the level of the players, identify their weaknesses, and consequently develop appropriate training programs to improve players’ performance [14]. The Illinois agility test is one of the most appropriate tests to measure the COD performance in team sports [15]. Using this test, coaches can also assess players’ progress over time and measure the effectiveness of training programs, such as speed, agility, and quickness training programs [15].

Improvement of COD performance depends almost exclusively on specific training [6], with stretching traditionally considered one such intervention [16]. Several authors have suggested that static stretching exercises exert an acute positive effect on muscle strength and power, with a beneficial outcome on COD performance [17,18,19,20]. On the other hand, static stretching before explosive movements may actually decrease performance [21,22]. Static stretching is often performed before exercise and actual games because it is believed that pre-exercise stretching decreases the risk of injury and improves performance [23]. However, numerous authors recommend dynamic stretching to improve performance [24], advocating that dynamic stretching should replace static stretching as the latter reduces soccer performance [24]. Some studies have shown that both static and dynamic stretching exerted no positive effect on either speed or COD performance [25]. Thus, for better training adaptation, perhaps static and dynamic stretching should both be performed [26,27].

However, the literature on combined stretching is conflicting. Some authors reported negative effects in vertical jump [28] and sprint performance [29,30], whereas others reported no effect on vertical jump [31,32], sprint, COD, and jump height performance [28,29]. That is why the effects of combined static and dynamics stretching techniques have presented unclear results, particularly for COD [33,34,35].

A limited number of studies have investigated the effect of stretching exercises on COD in soccer players [36]. In addition, it is important to know the effect of the order of each stretching combination on physical performance. No studies have examined the difference in effect between combined static–dynamic (static stretching followed by dynamic stretching) versus combined dynamic–static stretching (dynamic stretching followed by static stretching) on COD. Moreover, despite studies examining the effects of stretching on a variety of physical performance parameters, only a few crossover studies with repeated measurements have reported the same participants subjected to various stretching methods, with standardization in relation to the types of tests applied, intensity, and volume of stretching, allowing direct comparison of results, particularly among trained athletes [25]. Based on this, this study examined the effects of a warm-up exercises including five different types of stretching protocols on the COD in adolescent elite male soccer players. We hypothesized that a combination of static–dynamic stretching and/or dynamic–static stretching would be more advantageous than static stretching alone for young elite soccer players [37].

## 2. Materials and Methods

### 2.1. Participants

Sample size was calculated with a power analysis (G*Power V 3.1.9.6), and this indicated that seven participants were required. The calculation was based on a medium effect size (Cohen’s d = 0.50) for COD with an alpha level of 0.05, statistical power of 80%, one group with 5 measurements, for repeated measures, within subject analysis of variance (ANOVA). However, in an effort to enhance the study’s robustness and account for potential attrition, we ultimately recruited a slightly larger sample of participants, totaling 22. Twenty-two elite U-17 male soccer players voluntarily participated in the study. Four participants were excluded due to specific reasons: two were affected by musculoskeletal injuries, while two others declined participation for personal reasons. Ultimately, twelve male soccer players (age: 16.3 ± 0.3 years, height: 1.81 ± 0.10 m, body mass: 67.7 ± 7.2 kg) successfully completed all stretching protocols and performance assessments (Figure 1). Players were included when they voluntarily agreed to participate (by providing written consent) and were already registered with the soccer club. Players were excluded from this study if they were suffering from any current musculoskeletal injury, neuromuscular disorders, or any medical condition. All subjects played in a second national championship soccer division (Olympic Oued Ellil) team, had over eight years of soccer training experience, and actively participated in official games throughout the season. They trained sessions for 4 to 5 days a week (~90 min per session), and playing one game over the weekend.

### 2.2. Ethical Considerations

The researchers explained the details of the study to the players. Individual written consent was obtained from all participants and their parents, if applicable, after they had received both oral and written explanations of the experimental procedure and its possible risks and benefits. The study procedures were revised and approved by a local medical research Ethics Committee (CEUR17ISSEPKS 17-S2/2022), according to the national regulation, and conducted in accordance with the Declaration of Helsinki.

### 2.3. Experimental Design

The study was conducted over two weeks, during an official competitive soccer season (on March). To minimize circadian rhythm effects, all sessions were held in the evenings in the same temperature and humidity ranges [38,39]. Before definitive testing sessions, all subjects completed a 2-week familiarization period with the test, to minimize the potential learning effect, which could confound true study effects. The testing protocol involved COD testing. The experimental conditions were randomized over the two weeks (Figure 2):

Week 1:Session 1: no stretchingSession 2: static stretchingSession 3: dynamic stretching

Week 2:Session 4: no stretchingSession 5: combined static–dynamic stretchingSession 6: combined dynamic–static stretching

The subjects in the combined static–dynamic stretching performed the same actions, hence stretching identical muscles, as in the static stretching and dynamic stretching protocols, except they first performed the static stretching followed by the dynamic stretching protocol. This was reversed in the combined dynamic–static stretching.

The first and the second day of tests were conducted over two days, with 72 h separating each session. The no-stretching condition’s COD served as a control. The study was realized during November, as part of an official competitive season. To minimize the effects of circadian rhythm on performance, all sessions were conducted at the same time as the subjects’ regular training session (in the evenings) and within the same temperature and humidity ranges (23–25 °C and 38–42%, respectively).

### 2.4. Training Program: General Warm-Up and Stretching Protocols

The interventions were performed after a general 20 min general warm-up, consisting of a 10 min light jogging followed by a 10 min eight exercises session (Table 1).

After completing the general warm-up and pre-exercise intervention (or no intervention in the control condition) and a rest for 2 min, the subjects performed the COD. Each stretching condition included approximately 6 min of stretching intervention. The duration of stretching intervention for each combined stretching condition was 12 min. However, in the control condition, COD was performed after 2 min of recovery following the general warm-up without performing any stretching. Figure 3 gives a visual overview of the protocols (Table 1, Table 2 and Table 3).

The static stretching protocol consisted of 6 min of stretching focused on the lower body and 5 min of walking. After walking at a normal pace, players performed six static stretches. For each leg, each stretch was performed twice (Table 2) [40]. Before moving on to the next leg or the next stretch, each stretch was performed for 10 s at the point of mild discomfort, followed by 5 s of relaxation.

Dynamic stretching consisted of ten dynamic exercises ranging in intensity from moderate to high, which were performed for 6 min (Table 3). The subjects performed each dynamic exercise for a distance of 10 m, took a 10 s break, and then performed the same exercise again for 10 m as they came back to the starting line. Throughout the performance of the dynamic movements, subjects were constantly instructed to maintain proper form (such as a vertical torso, knees in front of chest, and feet on toes).

### 2.5. Testing Procedures: COD

Participants completed the Illinois agility test (COD), which involved running around cones and following a prescribed route [26]. Each stretching condition included approximately 6 min of stretching intervention, except for the combined stretching conditions, which involved 12 min of stretching. The control condition involved no stretching. Participants performed two attempts at each test, with a 2 min rest between trials, and the fastest trial was used. The dimensions and itinerary direction for the COD were in accordance with established methods [41,42,43,44,45]. The test involved four cones being placed to indicate the agility area (10 m long × 5 m wide). Another four cones were placed in the center and were spaced 3.3 m apart. A photocell system (Microgate, Bolzano, Italy) was placed at the start line and at the finish line. The gates were positioned 1.75 m apart and the height of the gates was set to 0.75 m. The player lies in the prone position 0.6 m behind the starting line [46]. Players were instructed to run as quickly as possible around the cones and not to cut over them. In addition, subjects were informed to follow the prescribed route throughout the whole trial. If a player failed to follow this protocol, the trial was stopped and reattempted after the required recovery period. Players performed two attempts at each test with 2 min rest between trials; the fastest trial was used for analysis.

### 2.6. Statistical Analysis

Data are presented as mean ± standard deviation. SPSS (version 27.0; SPSS Inc., Chicago, IL, USA) was used for statistical analysis and the level of significance was set at *p* < 0.05. The normality of the distributions was determined using the Shapiro–Wilk test. Moreover, sphericity and homogeneity were checked using Maluchly’s test and Levene’s test, respectively. One-way repeated measures analysis of variance (ANOVA) was used to assess the stretching warm-up condition related effects. Bonferroni-adjusted pairwise comparisons were used. Reliability (intraclass coefficient correlation (ICC)), and standard error of measurement (SEM), SEM%, coefficient of variation (CV%) and minimal detectable difference (MDD) were evaluated. The SEM% was obtained by dividing the resulting estimate of the SEM by the mean for the participants in all trials, then multiplying by 100 [46]. The CV% was calculated for each athlete and then averaged out across the team through the division of the standard deviation by the mean of COD (CV = 100 (SD/M)). The minimum detectable difference at the 95% CI (MDD) was calculated using the following equation: MDD = z × standard error of measurement × √2, where z is 1.96 and the standard error of measurement is SD√(1−ICC) [47,48,49]. Effect size was presented using partial eta squared (η_p_^2^), were 0.01 < η_p_^2^ < 0.06 is defined as a small effect, 0.06 < η_p_^2^ < 0.014 is defined as medium, and η_p_^2^ > 0.14 resembled a large effect [50].

## 3. Results

### 3.1. Statistical Power and Reliability

Statistical power for this analysis was 0.95 and partial eta squared was 0.14. In addition, the results showed that the tests were highly reliable (CI = 0.67–0.95; ICC = 0.86; average CV = 1.8%; average SEM = 0.11). Descriptive statistics (Mean ± SD; SEM, SEM% and MDD) for each condition are presented in Table 4.

### 3.2. Analysis with Repeated Measures

Repeated-measure ANOVA results revealed no significant effects on COD performance after the different stretching protocols (F = 1.73, *p* = 0.161, η_p_^2^ = 0.14). However, the Bonferroni adjustment for multiple comparisons (pairwise comparison) revealed that the dynamic stretching protocol resulted in faster COD times compared to static stretching (*p* < 0.01, η_p_^2^ = 0.66), as well as both the combined static–dynamic stretching (*p* < 0.05, η_p_^2^ = 0.56) and combined dynamic–static stretching (*p* < 0.01, η_p_^2^ = 0.73). Furthermore, there was no statistically significant difference between dynamic stretching and no stretching (*p* = 0.146, η_p_^2^ = 0.18) (Figure 4).

Nevertheless, no notable variations were observed among the remaining conditions (Table 4). In regard to the static stretching condition, while there was a slight decrease in mean COD performance (16.46 ± 0.37 s) compared to the no stretching condition (16.58 ± 0.96 s), this disparity did not reach statistical significance (*p* = 0.637, η_p_^2^ = 0.02). Likewise, the mean time for static stretching, when compared to the mean times of both the combined static–dynamic (16.53 ± 0.43 s) and dynamic–static stretching (16.58 ± 0.46 s) conditions, exhibited no significant differences (*p* = 0.500, η_p_^2^ = 0.04 and *p* = 0.124, η_p_^2^ = 0.20, respectively). This meant that the COD times in the combined static–dynamic as well as the combined dynamic–static stretching were comparable to the no-stretching level.

## 4. Discussion

To our knowledge, this is the first study to compare the acute effects of static, dynamic, combined static–dynamic and dynamic–static stretching protocols on the change of direction performance in youth soccer players. Collectively, the combination of statistical power, effect size, and various reliability metrics firmly established the high reliability of the tests we conducted. These quantitative indicators, supported by detailed descriptive statistics, lend substantial confidence to the integrity and validity of our research outcomes. Our findings align with prior studies that have examined the reliability of the COD test [51,52,53]. Stewart et al. [53] reported robust reliability (0.88 to 0.95) and low error rates (1.95–2.40%) in COD tests, including the Illinois and T-test. These tests exhibit strong correlations (0.84 to 0.89), making them suitable for assessing COD. Raya et al. [52] examined active males and identified reliable agility tests (T-Test, Illinois Agility Test) that highlight diverse movement planes. Hachana et al. [51] confirmed the high reliability (ICC = 0.96) and validity of the Illinois Agility Test for assessing COD in male athletes, with a stronger association with speed rather than acceleration or leg power.

Our analysis using repeated-measures ANOVA revealed intriguing outcomes. Although no overall significant effects on COD performance were observed across the various stretching protocols, closer examination through pairwise comparisons under the Bonferroni adjustment revealed noteworthy findings. Dynamic stretching displayed an advantage, leading to faster COD times when compared to static stretching (*p* < 0.01), as well as both combined static–dynamic stretching (*p* < 0.05) and combined dynamic–static stretching (*p* < 0.01). Thus, our data do not support the hypothesis that a combination of static–dynamic stretching and/or dynamic–static stretching is more advantageous than static stretching alone for young elite soccer players under the experimental conditions used in this study. However, as we also identified dynamic stretching as the best stretching protocol in the design of our experiment, our results do not warrant a clear rejection of the hypothesis either. Interestingly, no significant difference emerged between dynamic stretching and no stretching. This suggests that soccer players engaging in a twenty-minute general warm-up before competition, whether with or without dynamic stretching, might exhibit similar COD performance.

Although after the dynamic stretching protocol, no significant faster completion time compared to the no-stretching condition was observed, 7 of the 12 participating players achieved a better performance in COD when they integrated the dynamic stretching component within their warm-up protocol. Consequently, our data support the hypothesis that dynamic stretching is beneficial compared with static stretching for young elite soccer players. These results are in line with previous research [26,40]. The mechanisms through which dynamic stretching improves muscular performance can be attributed to its active contractile nature, resulting in increased blood flow to the muscles, increased muscle tissue and body temperature, and accelerated nerve conduction velocity [54]. This process potentially improves motor control and heightened nerve receptor sensitivity, allowing faster and more forceful muscle contractions [54,55]. Furthermore, dynamic stretching can reduce muscle stiffness and induce positive changes in energy system metabolism [54]. Faster COD times obtained after dynamic stretching [56], as an active contractile process, could be also associated with higher post-activation potentiation in the stretched muscle, as well as the absence of stretch induced shortfalls [57], a nervous system stimulation, and/or decreased antagonist muscle inhibition [58,59]. Based on the current studies, dynamic stretching may result in greater force production through post-activation potentiation and optimal muscle temperature, resulting in faster speed.

However, the static stretching and combined stretching protocols did not yield performance improvements compared to the no-stretching condition. These findings suggest caution in using static or combined stretching immediately before competitions or training sessions that require high power output. To date, very few authors have investigated the effect of static stretching on the COD of soccer players. The effect of static stretching on athletic test performance in soccer players is either detrimental or not detrimental to performance, but not beneficial [24,25,60]. Our findings are in accordance with prior studies, further emphasizing the adverse impact of static stretching on muscle performance [36,61,62]. Comparing the effects of static stretching to no stretching, our results demonstrated no statistically significant differences. Similarly, existing evidence indicates that the immediate impact of static stretching and no stretching is not significantly different [36]. However, a static stretching warm-up decreased performance in COD maneuvers on a 20 m zigzag course, while not negatively affecting the 20 m sprint times [36]. Conversely, Amiri-Khorasani et al. [26] highlighted a negative effect of regular static stretching, often included in soccer pretraining and pre-competition warm-ups, on subsequent COD performance when compared to no stretching. On the other hand, Little et al. [36] showed that for three of the four measures used (countermovement vertical jump, stationary 10 m sprint, flying 20 m sprint, and agility performance), there was no difference between static and no-stretch warm-up protocols, whereas in the 20 m maximal-speed test the static-stretch protocol produced significantly faster runs than the no-stretch protocol.

Another pertinent study aimed to compare the effects of various durations of static stretching followed by dynamic stretching on repeated sprint ability (RSA) and COD, yielding further insights into the impact of the absence of stretching compared to static stretching conditions, particularly when combined with dynamic stretching. The study revealed that the duration of static stretching exerted a positive influence on flexibility. Sit-and-reach test scores were 36.3% and 85.6% higher for the 60 s and 90 s static stretching conditions, respectively, compared to the 30 s condition (*p* ≤ 0.001), indicating substantial improvements in flexibility. However, despite these noteworthy gains in flexibility, there were no significant differences observed in RSA and COD performance across the three stretching conditions. From these results, it appears that the intensity and duration of the introduced warm-up protocols could explain the variance observed in comparison with our findings. Furthermore, it has been suggested that the increase in time required to complete the COD after static stretching could be attributed to alterations in the mechanical properties of the musculotendon unit or changes in neural activation that may lead to a decrease in force. In particular, research suggests that a regimen of static stretching may well provide acute inhibition of maximal force production by the stretched muscle [36].

Regarding combined stretching (combined static–dynamic stretching or combined dynamic–static stretching), our results showed that there were no significant differences between the two protocols and between the combined stretching and static stretching or no stretching conditions. In addition, COD performances were significantly better after dynamic stretching than after both the combined stretching protocols. It seems that the combination of static stretching and dynamic stretching may have counterbalanced the possible positive (i.e., dynamic stretching) and negative (i.e., static stretching) effects. The present results did not show significant impairments in COD time associated with prior static stretching or dynamic stretching when compared to no stretching.

However, static stretching following dynamic stretching may neutralize the positive effect of dynamic stretching. These findings can be explained by the fact that, during static stretching and other stretching methods, such as combined static stretching and combined dynamic stretching, muscles are stretched to excessive tension for a prolonged period, probably inducing an inverse stretch reflex by the Golgi tendons if excessive tension is imparted to the muscle [63]. These results contrast a study [24] examining the effect of combined stretching on 20 m sprint: any stretching protocol, followed by dynamic stretching, increased acceleration and speed. The last result can be explained by a determined novel approach, “COD deficit”, which represents the difference in speed between linear sprint and COD measurements of similar distance [64] or the additional time that COD requires when compared with a linear sprint over an equivalent distance [65]. In fact, in young soccer players (U20), better performances in speed tests are not necessarily related to better performances in specific COD [6].

Rosenbaum and Hennig [66] reported that additional physical activity after static stretching attenuated the decreases in peak force, rate of force development (RFD), relaxation rate, and electromyography amplitudes of the Achilles tendon reflex. Extra muscle activity after stretching may have reversed any decrease in muscular compliance and associated decreased neural drive initiated by stretching. Thereby, the effects of stretching on COD performance appear to vary depending on the method of stretching used [54]. These findings are important in choosing the most efficient mode of stretching to include in a warm-up protocol to produce the fastest COD times for soccer players.

On the other hand, closed-skill activities, such as sprinting, COD, and jumping, focus on improving specific physical attributes such as speed, agility, and power [67]. These qualities help players develop their muscular strength, coordination, and proprioception, which are all crucial for executing quick and precise movements during a game. These skills are particularly important in soccer, where players frequently need to change direction rapidly, accelerate, and jump to reach the ball or avoid opponents [68]. The study of the relationship between COD and straight running speed and vertical jump tests is of a high importance, as it might explain, at least in part, whether running speed and vertical jump are determinant factors for COD [13]. A well-designed warm-up should prime the muscles for the forthcoming physical exertion, without causing excessive fatigue [69]. A recommended minimum duration of around 10 min is generally acknowledged to yield optimal benefits [70].

## 5. Practical Applications

The results of this study provide evidence that the short-term effects of combined static–dynamic or dynamic–static stretching does not appear to have an adverse effect on COD performance. However, the evidence presented highlights the efficacy of dynamic stretching as a means to enhance COD ability during warm-up regimens. The practical implications of these findings hold significance for coaches, trainers, and athletes. Therefore, coaches can consider integrating dynamic stretching protocols tailored to the athletes’ specific needs and the demands of the ensuing activity. However, the cautionary note against utilizing static or combined stretching immediately before competitions and training sessions emphasizes the importance of optimizing warm-up strategies to ensure maximal performance gains.

Despite the valuable insights provided by this study, a number of limitations warrant consideration. First, the sample size was relatively small, consisting of only twelve male soccer players. This could potentially limit the generalization of the findings to a broader population. Secondly, the study only examined acute effects immediately following the stretching protocols, without investigating potential long-term adaptations. Additionally, the specific dynamic stretching exercises employed in the study might not encompass the entirety of dynamic stretching practices commonly used. Furthermore, the study focused solely on male youth soccer players, and the applicability of the results to other age groups or genders remains unexplored.

Building upon this study’s findings, several possibilities for future research may be explored. Investigating the long-term effects of dynamic stretching on COD ability could offer insights into its potential role in enhancing athletic performance over time. Exploring the mechanisms underlying the observed improvements in COD after dynamic stretching could deepen our understanding of the physiological processes at play. Furthermore, extending the investigation to encompass a wider range of athletes, including different age groups and genders, would enhance the applicability and generalization of the findings. Lastly, a comparative analysis of various dynamic stretching routines and their respective impacts on COD ability could contribute to refining and customizing warm-up protocols in different athletic contexts.

## 6. Conclusions

In practical terms, our results suggest that dynamic stretching exercises hold promise to enhance COD performance in youth male soccer players during warm-up routines. However, the “no-stretching” condition had comparable COD performance to dynamic stretching and exhibited no significant differences when compared to the other stretching protocols tested. This implies that, at least for the measure of COD performance, no stretching did not show a notable detrimental effect compared to the stretching protocols studied. Accordingly, coaches, trainers, and physical educators are encouraged to consider incorporating dynamic stretching into pre-activity warm-ups, particularly when demanding a high power output. Meanwhile, our findings advise against the use of static or combined stretching in close proximity to competitions or training sessions, as they did not show performance benefits compared to no stretching. In conclusion, this study contributes insights into optimizing warm-up strategies for youth male soccer players. By discerning the advantages of dynamic stretching and highlighting the limitations of static and combined stretching, our findings provide actionable guidance to enhance performance preparation and training regimens.

## Figures and Tables

**Figure 1 sports-11-00165-f001:**
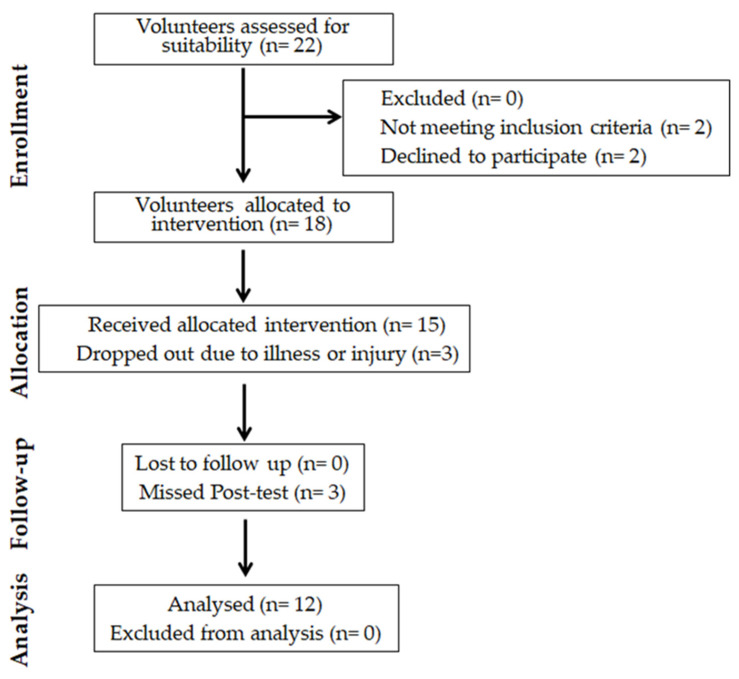
The diagram includes detailed information on the interventions received.

**Figure 2 sports-11-00165-f002:**
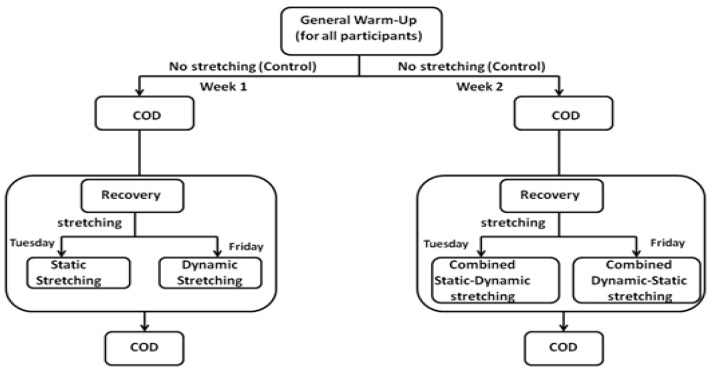
Schematic representation of the experimental protocol.

**Figure 3 sports-11-00165-f003:**
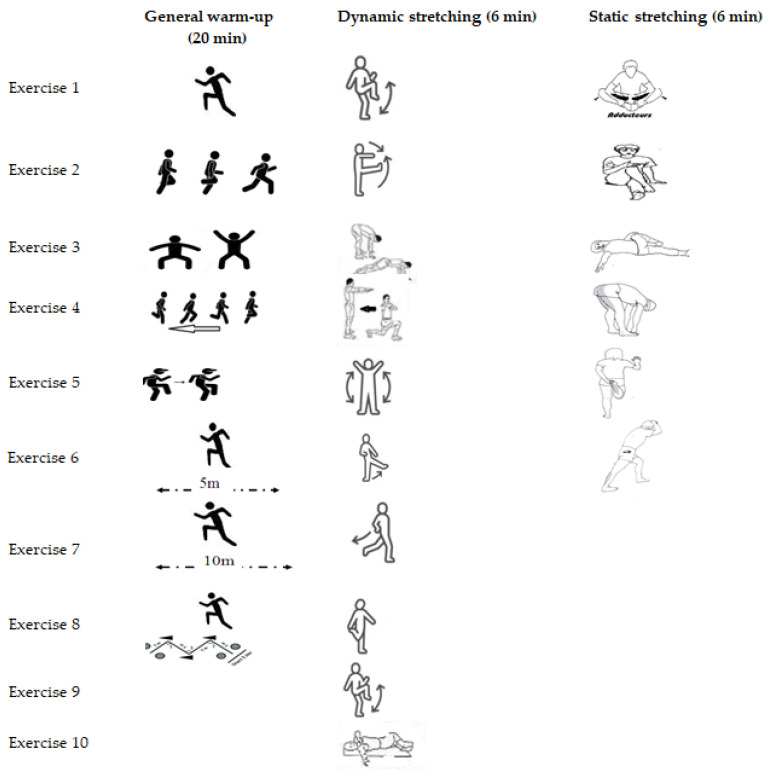
Training Program: General warm-up and stretching protocols.

**Figure 4 sports-11-00165-f004:**
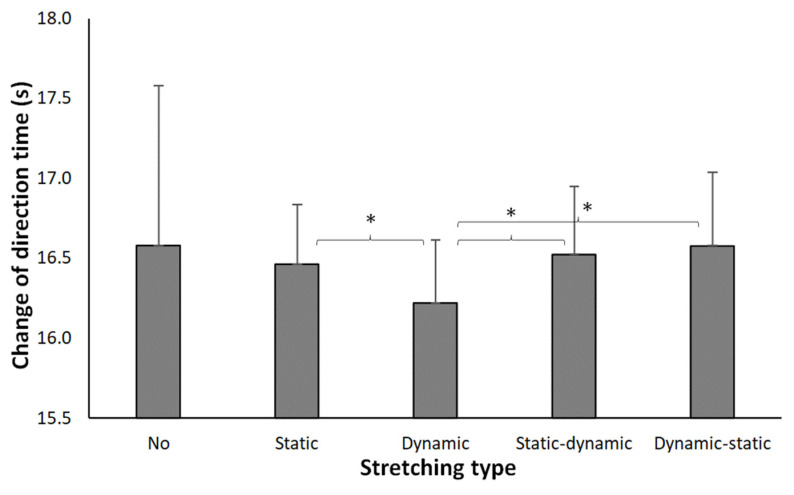
Mean (SD) COD times after each of the stretching protocols. * indicates a significant difference in time between these two protocols at a *p* < 0.05 level. No: no stretching.

**Table 1 sports-11-00165-t001:** Details of general warm-up performed by participants.

Warm-up run (jogging) (10 min): All participants completed 800 m at a low intensity for 5 min on the running track, followed by a submaximal aerobic warm-up consisting of linear running of progressive intensity for 5 min. Exercise 1: Light running and varied movements, including 1 min of moderate walking and light jogging, 1 min of sidestepping and back jogging, and 1 min of further jogging (3 min) Exercise 2: Lateral movement: side-to-side actions used to change direction (1 min) Exercise 3: Backward running with change of direction (1 min) Exercise 4: Walking forward with knee elevation to the chest and right trunk (1 min) Exercise 5: Running in a straight line over a distance of 5 m, repeated twice, with 10 s recovery (1 min) Exercise 6: Running in a straight line over a distance of 10 m, repeated twice, with 10 s recovery (1 min) Exercise 7: Running in a zigzag form over a distance of 20 m, repeated twice, with 10 s recovery (2 min)

**Table 2 sports-11-00165-t002:** Static stretching exercises [40].

Static Stretching Exercises (6 min Total)	Time (s)	Sets
1. Adductor stretch. In the seated position with an erect spine, touch soles of feet together, bend knees, and allow knees to drop.	30	2 Total
2. Modified hurdlers stretch. In a seated position with one leg straight, place the other leg on the inside of the straight leg and reach forward.	30	2 Total (1 per leg)
3. External hip rotator stretch. In a supine position, bring either ankle to the opposite knee and cross one leg over the other, forming a figure four position, and flex both hips to or past 908 by pulling on the uncrossed leg.	30	2 Total (1 per leg)
4. Bent-over toe raise. From a standing position with the heel of one foot slightly in front of the toes of the other foot, dorsiflexion front foot towards shin while leaning downward with upper body.	30	2 Total
5. Quadriceps stretch. In the standing position with an erect spine, bend one knee and bring heel towards buttocks while holding the foot with one hand.	30	2 Total (1 per leg)
6. Calf stretch. In a standing position with feet staggered about 2 or 3 feet from a wall, lean against the wall with both hands, keeping the back leg straight and the front leg slightly bent.	30	2 Total (1 per leg)

**Table 3 sports-11-00165-t003:** Dynamic stretching exercises [40].

Dynamic Stretching Exercises *†	Repetitions
1. High-knee walk. While walking, lift knee towards chest, raise body on toes, and swing alternating arms.	2 0
2. Straight-leg march. While walking with both arms extended in front of body, lift one extended leg towards hands then return to starting position before repeating with other leg.	10 per side
3. Hand walk. With hands and feet on the ground and limbs extended, walk feet towards hands while keeping legs extended, then walk hands forward while keeping limbs extended.	5
4. Supine knee rocking	2 0
5. Backward lunge. Move backwards by reaching each leg as far back as possible.	10
6. High-knee skip. While skipping, emphasize height, high knee lift, and arm action.	2 0
7. Lateral shuffle. Move laterally quickly without crossing feet (Carioca).	2
8. Back pedal. While keeping feet under hips, take small steps to move backwards rapidly.	10 CW/CCW
9. Heel-ups. Rapidly kick heels towards buttocks while moving forward.	2 0
10. High-knee run. Emphasize knee lift and arm swing while moving forward quickly.	2 0

* CW = clockwise; CCW = counterclockwise. † Total time: approximately 6 min. Carioca was performed twice (back and forth) with a 5 m distance.

**Table 4 sports-11-00165-t004:** Descriptive statistics (Mean ± SD) for the mean COD performance following each of the stretching protocols.

Condition	No-Stretching	Static	Dynamic	Static–Dynamic	Dynamic–Static
Mean ± SD	16.58 ± 0.96	16.46 ± 0.37 **	16.22 ± 0.39	16.53 ± 0.43 *	16.58 ± 0.46 **
SEM	0.15	0.17	0.14	0.20	0.13
SEM%	0.90	1.03	0.86	1.20	0.78
MDD	0.30	0.34	0.27	0.39	0.26

*: significantly different from dynamic *: *p* < 0.05); **: *p* < 0.01. SEM: standard error of measurement; SEM%: percent of SEM to the mean; MDD: minimum detectable difference.

## Data Availability

The raw data supporting the conclusions of this article will be made available by the authors without undue reservation.

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
