# Peer review of "Acute Effect of Four Stretching Protocols on Change of Direction in U-17 Male Soccer Players"

_sports, 2023, doi:10.3390/sports11090165_

Round 1
Reviewer 1 Report
Dear authors,
Your manuscript presents comparisons of different different stretching methods and no stretching in youth soccer players. Indeed, it is a relevant study to the state of art. However, there are some parts of the manuscript that can be improved.
For instance, several format issues were found in the manuscript. Then, hypothesis of the study should be added in introduction. Methods should present a sub-section with the study design. In participants section, eligibility criteria should be added. General warm-up should better describe its intensity and time/reps for each stretching exercise, so other researchers can replicate the same protocol (this is the major issue of the study).
The second major issue is related with lines 169-172. Such results are unusual. When ANOVA shows no significant results, a post hoc analysis also does not provide a significant result, but this was not the case. Can the authors please confirm the results? Moreover, since authors present them in the figure 2, it is suggested to provide more inferential statistics for all comparisons. Please add a new table or add them in text.
Discussion can be improved by elaboratin more on no stretching vs. static stretching conditions because it would be interesting to analyse if we should include or not dynamic stretching to improve COD. There should be some reference on this topic in conclusions. Lastly, limitations, practical implications and future directions are missing.
More comments were made in attachement. Please note that I use microsoft edge for comments.
Best regards

Minor editing of English language required.
Author Response
See pdf file

Reviewer 2 Report
Dear Authors,
The aim of this study was to compare the effects of different stretching methods on change of direction ability (COD) in soccer players. It is an interesting manuscript with a strong research gap, methodological procedures and results, however some sections need revisions.
Please, consider the following point-by-point revisions in the attachment.
Good work!

Author Response
see pdf file

Reviewer 3 Report
This study is generally interesting, but in has many flaws, which should be addressed. The study group is very small. Was the power calculation calculated? If not it should be explained in details.
There is no deep description of the study group - what were the inclusion/exclusion criteria?
The presented study protocol does not allow for the assessment of ICC. Reliability can not be calculated from the presented methodology. Therefore, presented in the results section the ICC is not justified and is a mistake. This indicates the wrong statistical methods used in the calculations.
What is the measurement repeatability of the change of direction test based on the available literature? What was the standard error of measurement SEM and minimum detectable difference MDD? With such a small group, small differences in speed can influence the results. Data in the results section should be supplemented with MDD, SEM, CV, confidence interval CI.
The results are presented too generally in figure 2. Moreover, the ICC is calculated incorrectly. The results require a complete revison
Without improving the methodology and results, the discussion and conclusions seem to be over-interpreted.
Author Response
see pdf file

Reviewer 4 Report
Thank you for the opportunity to review this manuscript.
The article is interesting, but needs major improvements to be published.
Recommendations.
Highlight the novelty of the study compared to the numerous previous studies that focused on agility and changes in directives in football players.
In Methods, only one test was applied which was adapted without explaining why this adaptation was made. With a single applied test it is difficult to highlight the novelty aspects of the study and the relevance of the implemented experimental training program.
Unfortunately, the Results section is irrelevant, it does not contain tables with data and statistical processing.
The Discussion section is too general and does not refer to the relevant results of the present study, but only reviews certain results from previous studies.
The conclusions are not correlated with the main results of the study and are too general.
Author Response
See pdf file

Round 2
Reviewer 1 Report
Dear authors,
Congratulations on your work. All changes and answers to my comments were very well addressed. Still, I have minor comments to improve quality of your work:
-in the abstract, please check the writing for a better transition between sentences, namely, between background and aim of the study;
-L47-51, When authors mentioned several studies, it is expected that more than one study is cited in the same sentence. Thus, please add more references or change the writing by replacing several studies per several authors;
-section 2.1 - this section can be merged with participants section and "participants" should be the title;
-L102-105, please state the test family, statistical test and type of power analysis used;
- The rectangles of figure 1 are not well placed. Please check;
-References are not following Sports guidelines.
Best regards
Minor editing of English language required
Author Response
Comments and Suggestions for Authors
Dear authors,
Congratulations on your work. All changes and answers to my comments were very well addressed. Still, I have minor comments to improve quality of your work:
- Point 1: -in the abstract, please check the writing for a better transition between sentences, namely, between background and aim of the study;
Authors Response 1:
According to the reviewer’s suggestion, in the “Abstract” section, the transition between sentences was improved between background and aim of the study. Therefore, the text was changed as follows:
“Background: The ability to rapidly change direction while sprinting is a desirable athletic skill in soccer. Enhancing change of direction (COD) performance depends almost exclusively on specific training, with stretching traditionally considered one such intervention. However, the comparative impact of diverse stretching methods on COD in soccer players remains an area of interest. Therefore, this study aimed to compare the effects of different stretching methods on COD ability in soccer players.”
- Point 2: -L47-51, When authors mentioned several studies, it is expected that more than one study is cited in the same sentence. Thus, please add more references or change the writing by replacing several studies per several authors;
Authors Response 2:
We appreciate the reviewer’s insightful suggestion and as suggested, the writing was changed by replacing “several studies” per “several authors”
- Point 3: -section 2.1 - this section can be merged with participants section and "participants" should be the title;
Authors Response 3:
In accordance with the suggestion of the reviewer, the “-section 2.1” was merged with “Participants” section and "Participants" is now the title.
- Point 4: -L102-105, please state the test family, statistical test and type of power analysis used;
Authors Response 4:
The test family: F tests;
Statistical test: ANOVA; repeated measures; within factors
Type of power analysis: A priori: Compute required sample size-given a, power, and effect size.
In accordance with the suggestion of the reviewer, the test family, statistical test and type of power analysis used was added to the revised manuscript as follows:
“Sample size was calculated with a power analysis (G*Power V 3.1.9.6) and it indi-cated that 7 participants were required. The calculation was based on a medium effect size (Cohen’s d = 0.50) for COD an alpha level of 0.05, statistical power of 80%, one group with 5 measurements, for repeated measures, within subject analysis of variance (ANOVA). However, in an effort to enhance the study's robustness and account for potential attrition, we ultimately recruited a slightly larger sample of participants, totaling 22.”
- Point 5: - The rectangles of figure 1 are not well placed. Please check;
Authors Response 5:
Figure 1 was checked and corrected as suggested.
- Point 6: -References are not following Sports guidelines.
Authors Response 6:
In accordance with the suggestion of the reviewer, the references were done following Sports guidelines.
We most sincerely thank Reviewer 1 for taking the time to review our manuscript and we greatly appreciate the reviewers’ thoughtful comments and hope they will agree that the revised version of our manuscript has much improved.
Reviewer 2 Report
Dear Authors,
Overall, the authors improved the article extensively after peer review.
I would warn you to refer to the abbreviation in full only once throughout the manuscript (examples, lines 20 and 22 in the abstract; lines 43, 46 and 53 in the introduction).
Figure 1 is unformatted (as are most of the tables and articles).
References to support the materials and methods are missing.
Add the study's limitations and future prospects to the "practical applications" sub-chapter.
With best regards,
José Eduardo Teixeira.
Author Response
Comments and Suggestions for Authors
Dear Authors,
Overall, the authors improved the article extensively after peer review.
- Point 1: I would warn you to refer to the abbreviation in full only once throughout the manuscript (examples, lines 20 and 22 in the abstract; lines 43, 46 and 53 in the introduction).
Authors Response 1:
We appreciate the reviewer’s insightful suggestion and as suggested, the abbreviations were referred in full only once throughout the revised manuscript.
- Point 2: Figure 1 is unformatted (as are most of the tables and articles).
Authors Response 2:
Figure 1 and all the other figures and tables were checked and corrected as suggested.
- Point 3: References to support the materials and methods are missing.
Authors Response 3:
In accordance with the suggestion of the reviewer, some references was added in the “Materials and Methods” section to support the materials and methods.
The references added to this section:
- Atkinson, G.; Reilly, T.; Circadian variation in sports performance. Sports Med. 1996, 21(4):292-312. doi: 10.2165/00007256-199621040-00005.
- Chtourou, H.; Hammouda, O.; Souissi, H.; Chamari, K.; Chaouachi, A.; Souissi, N. The effect of Ramadan fasting on physical performances, mood state and perceived exertion in young footballers. Asian J. Sports Med. 2011, 2: 177–185. doi: 5812/asjsm.34757
- Pérez-Ifrán P, Rial M.;, Brini S.;, Calleja-González, J.; Del Rosso S.;, Boullosa D.; and Benítez-Flores, S. Change of Direction Performance and its Physical Determinants Among Young Basketball Male Players. Hum. Kinetics 2023, Volume 85 (2022): Issue 1 DOI: https://doi.org/10.2478/hukin-2022-0107.
- Point 4: Add the study's limitations and future prospects to the "practical applications" sub-chapter.
Authors Response 4:
In accordance with the suggestion of the reviewer, the study's future prospects and limitations were added to the "Practical Applications" sub-chapter.
Once again, we sincerely appreciate your feedback, which will undoubtedly help us strengthen the clarity and impact of our manuscript. We look forward to incorporating your suggestions and resubmitting an improved version of this manuscript.
Reviewer 3 Report
Authors have addressed all my comments
Author Response
Thank you for the review
Reviewer 4 Report
The authors improved the manuscript.
Author Response
Thank you for the review